# Green Fences for Buenos Aires: Implementing Green Infrastructure for (More than) Air Quality

**María del Carmen Redondo Bermúdez [1],\*, Juan Miguel Kanai [2] , Janice Astbury [2], Verónica Fabio [3] and Anna Jorgensen [1]**

[1] Department of Landscape Architecture, Faculty of Social Sciences, The University of Sheffield, Sheffield S10 2TN, UK; a.jorgensen@sheffield.ac.uk

[2] Department of Geography, Faculty of Social Sciences, The University of Sheffield, Sheffield S10 2TN, UK; miguel.kanai@sheffield.ac.uk (J.M.K.); janice.astbury@gmail.com (J.A.)

[3] Faculty of Architecture, Design and Urbanism, University of Buenos Aires, Intendente Güiraldes 2160, Buenos Aires C1428EGA, Argentina; veronica.fabio@fadu.uba.ar

\* Correspondence: maria.redondo@sheffield.ac.uk

**Abstract:** Schoolyards in North America and Europe are increasingly using green fences as one measure to protect vulnerable populations from localised air pollution. This paper assesses the possibilities and limits for mobilising this format of site-specific green infrastructure in cities in low- and middle-income countries beset by air pollution and multiple other socio-environmental challenges, and particularly questions the definition of green fences as a green infrastructure for air quality (GI4AQ). We applied several qualitative and action research methods to the question of green fence implementation in Buenos Aires, Argentina—a Latin American city with weak air-quality policies, limited green infrastructure, and little experience with nature-based solutions. Firstly, we conducted a literature review of the role that urban vegetation and ecosystem services may play in AQ policy and the implementation barriers to such approaches globally and in the city. Secondly, we planned, designed, constructed, maintained, and evaluated a pilot green fence in a school playground. Thirdly, we carried out supplementary interviews with stakeholders and expert informants and compiled project members' narratives to respectively characterise the barriers that the project encountered and delineate its attributes based on the associated actions that we took to overcome such barriers to implementation and complete the pilot. Our findings identify multiple barriers across seven known categories (institutional, engagement, political, socio-cultural, built environment and natural landscape, knowledge base and financial) and highlight examples not previously considered in the extant international literature. Furthermore, learning from this experience, the paper proposes an expanded model of green infrastructure for air quality *plus multidimensional co-benefits* (GI4AQ+) to increase implementation chances by attending to local needs and priorities.

**Keywords:** action research; nature-based solutions; air quality; green infrastructure; urban environmental policy; Latin America

## 1. Introduction

Air pollution poses substantial threats to human health, making air quality (AQ) management a key policy area. In 2016, 4.2 million people died prematurely due to illnesses related to poor ambient AQ globally. According to the World Health Organization (WHO) [1], over 90% of these deaths occurred in low- and middle-income countries. The United Nations Environment Programme (UNEP) lists weak AQ regulations, lax vehicle emission standards, and remaining coal-burning power plants as potential reasons for disproportionate air pollution-linked deaths amongst the poorest populations [2]. Poor AQ compromises children's health in particular. They are smaller and breathe more air per unit body weight; therefore, the effects of pollutants are amplified [3]. Additionally, children's

lungs are still in development, and early exposure to air pollution has a negative association with lung function later in their lives [4,5]. Air pollution also renders children more susceptible to developing asthma and other respiratory problems [6,7]. Other consequences include cognitive impairment, depression, and anxiety symptoms [8].

Whilst efforts to abate urban air pollution date back decades, the study and practical use of GI to improve AQ is a relatively new field. The scientific literature has surged since 2017 [9]. Multiple GI formats can potentially mitigate air pollution. These include urban forests, green walls, green roofs, tree-lined streets, and green fences. The latter are gaining recognition as a valid approach to mitigate urgent air pollution problems in urban settings across the USA, Europe, and the UK. Moreover, Hewitt et al. [10] recently coined the term green infrastructure for urban air quality (GI4AQ) to designate GI that has been specifically and effectively designed to protect specific sites from air pollutants. The authors assign a 'third order' rank of priority to such GI, behind the overall reduction and removal of pollution sources from urban environments where vulnerable populations live and spend time. They conceptualise this as a Reduce-Extend-Protect model. More broadly, the consensus on GI4AQ includes the following three mitigation mechanisms: (1) dispersion of pollutant gases, such as nitrogen dioxide ($NO_2$), and of particulate matter (PM); (2) deposition of PM on plants' external structures; (3) distance elongation between air pollution sources and receptors [11]. Further guidance has been given by governmental institutions, such as the United States Environmental Protection Agency (US EPA) [12] and the Greater London Authority [13], demonstrating the case for GI to be incorporated into AQ policy and practice. In fact, the US EPA [12] specifically recommends using green fences to mitigate air pollution in school campuses exposed to vehicular traffic.

The GI4AQ model is not the sole approach. The potential for urban vegetation to contribute to AQ management is of interest for proponents of nature-based solutions (NbS), which are defined as a set of integrative principles and actions inspired by natural processes to address environmental, economic, and social challenges through ecosystem-based services [14,15]. The NbS perspective indicates that there is much to be gained by envisioning GI as a component of multi-pronged, multi-purpose urban sustainability strategies rather than isolated interventions with limited site impact. In fact, the European Union (EU), one of the major promoters of NbS, define GI as "a strategically planned *network* of natural and semi-natural areas with other environmental features designed and managed to deliver a wide range of ecosystem services" (emphasis added) [16] (p. 3). Calfapietra [17] (pp. 144–154) reviewed several EU-funded research and innovation projects seeking to ascertain NbS' role in AQ management. Many of these include related objectives of 'co-benefits', such as microclimate regulation and thermal comfort, human health and wellbeing, and urban biodiversity improvements.

This paper makes the case that green fences need not be limited to a single AQ function. Instead, they could contribute to a transformative urban environmental agenda. Strategically designed and curated green fences in schoolyards can produce multiple socio-environmental gains in the short run in synergy with longer-term educational and awareness-raising actions. These will address the root causes of air pollution and other forms of environmental degradation in cities. We call this the green infrastructure for air quality *plus socio-environmental co-benefits* (GI4AQ+) model for green fences. GI4AQ+ can be applied in schools to protect children at the place where they spend one third of their days, but green fences can also be adopted by those aiming to provide co-benefits beyond the reduction of air pollutant concentrations on site. Considering green fences as an urban NbS also requires grappling with their implementation potential and barriers in real-life contexts. Chatzimentor et al. [18] argue the need for more socially oriented NbS research and find that implementation constitutes the most recommended topic for future studies (in 62 out of 196 scientific publications from the EU and UK). Global scalability remains a challenge for this literature, however, and there is practically no specific discussions on green-fence formats.

There is a growing global consensus to incorporate GI and NbS into the urban policy toolbox, with endorsement by institutions such as the United Nations [19]. Nevertheless, implementation varies markedly across nations and cities, which have developed comprehensive schemes through regulatory reforms, policy innovation, and creative incentives. Research coverage is also uneven. This is particularly the case for site-specific GI formats (similar to green fences) such as green roofs and green walls, for which Liberalesso et al. [20] identified six mechanisms worldwide to support adoption. These include tax reductions, financing, construction permits, sustainability certifications, legal mandates, and agile administrative processes. However, these mechanisms cluster in Europe, the US, and Canada, whilst South America and Asia have considerably fewer examples. This justifies the need for further studies from cities in underrepresented regions, which could also address gaps in the broader field of GI research; studies using related concepts such as ecosystem services, NbS, and natural capital are mostly produced in Europe, the US, and Canada [21]. In fact, these three regions had more than 2000 publications each by 2017, followed by Asia (mainly China), with more than 1000 publications about GI/ecosystem services. In comparison, Latin America and the Caribbean, this paper's world region of interest, had under 500 publications [22]. In Latin America, GI adoption has focused on biodiversity conservation, climate change mitigation, and recreation and health. Regarding AQ, studies mostly link with climate change research. Air quality benefits are explored in relation to large landscape interventions, such as urban forests in Chile [23] and Mexico [24,25]. Some countries, such as Brazil, Mexico, Peru, Colombia, and Argentina, have introduced the GI concept in the climate change/air quality context. However, Vásquez et al. [21] argue that GI adoption in these countries' current policies is minimal, with insufficient implementation by planners and practitioners.

The low coverage of GI and AQ mitigation research in low- and middle-income countries, such as Latin American countries, does not reflect the fact that these nations have vulnerable populations at high risk from air pollution. Furthermore, the assessment of barriers to implementation for green fences in Latin American cities cannot easily build on policies underway. In Buenos Aires, for example, no such scheme existed previous to our pilot. To close these knowledge gaps, mitigate a serious health threat, and contribute to the broader NbS agenda, this paper discusses the actual implementation experience of a pioneer green fence and development of the GI4AQ+ model. We first list the range of qualitative and action research methods used and then present and discuss our findings. Our main concern is implementation potential in the city and Latin American region, and thus our analytical focus lies in the identification of barriers and mitigating strategies.

## 2. Methods

This study combines (i) a literature review of key concepts related to GI implementation and its use in AQ policy with (ii) a case study of a green fence built through researcher-initiated action research. We first carried out a preparatory literature review on global barriers to GI implementation and on the current approach of Buenos Aires with respect to GI and AQ. We then engaged in the process of building a green fence in a schoolyard (playground) in Buenos Aires, which we used as the basis for an explanatory case study of barriers to GI implementation in the city and potential strategies to overcome them. From here on, we refer to the pilot as the GF-BA project (Green Fence in Buenos Aires project). GF-BA was a researcher-initiated co-production project that involved planning, designing, constructing, maintaining, and assessing a pilot green fence in a school playground; this was the first of its kind in Buenos Aires. Figure 1 presents a summary of our study's research design.

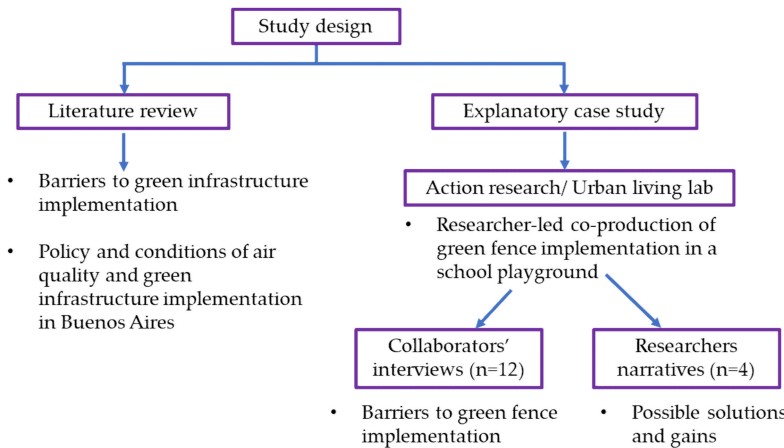

**Figure 1.** Research design of the study.

The GF-BA project involved multiple city government units and various social stakeholders in a collaborative network (see Figure 2), focused on obtaining practical insight on green fence implementation [26,27]. The GF-BA incorporated urban living lab (ULL) principles from the outset, particularly in what concerns the green fence's experimental character at the complex intersection between environmental science and sustainability innovations, and the active involvement of various stakeholders and community users beyond researchers and policy makers [28–30].

**Figure 2.** Overview of actors involved in the GF-BA project.

### 2.1. Literature Review

Our literature review on the international current state of the art regarding barriers to green infrastructure implementation aimed to identify analytical categories to structure our primary data collection in Buenos Aires. This was not limited to the AQ field due to the very limited number of publications addressing this issue within this field and the fact that barriers to other GI formats may also apply to green fences. The international typology of barriers that we developed also helped us identify barriers specific to the Buenos Aires context and to categorise the various actions taken to lead the GF-BA project to fruition. This also required in-depth understanding of the local context in terms of AQ and green infrastructure policy, for which we carried out an additional review of scientific sources and policy documents in both English and Spanish.

### 2.2. Explanatory Case Study Based on Action Research—The GF-BA Project

Our data collection pivoted around action research. Action research refers to a repeated cycle of action and reflection. It is anchored in an ethos of transformative interventions, whose beneficiaries lead or participate actively in their creation [31] (pp. 90–95). We selected action research for its potential to effect change by generating solutions to practical problems while empowering people to engage with activities on the ground and participate actively in the research [32] (pp. 1–24). This makes action research particularly suitable to advance NbS research in settings without robust policy mechanisms for GI provision, where researchers are unable to study pre-existing examples. Previous to the GF-BA pilot, there was no provision of vegetation to mitigate air pollution in Buenos Aires. We started from the assumption that action research projects can support participants (researchers and practitioners/stakeholders) in co-creating GI whilst also observing and analysing its effects. Ideally, participants will work together in a collaborative way, including contributing ideas, trying things out, sharing experiences, generating knowledge, adjusting course within systematic cycles of action, and reflection [33] (p. 5). Therefore, as problems emerged and were dealt with in an iterative 'learning by doing' approach, action research provided us with 'unique access to insider knowledge' [34] (p. 5), which we leveraged to ascertain latent barriers to GI implementation.

In November 2019, we installed the pilot green fence at a school in the western neighbourhood of Floresta (see Figure 3). This location was selected at the end of a year-long process of liaising with multiple stakeholders and gatekeepers. We visited multiple city-run schools until finding the right fit. There were requirements from the city government's Green Schools (GS) unit and from the GF-BA research team. Most notably, for GS, the school needed to be affiliated with their programme, have an 'Environmental Reference' teacher and track record of a school garden. For the GF-BA research team, the school needed to have measurable pollution levels (i.e., high traffic flow) and have an open schoolyard facing the street, where a landscape intervention was possible.

The pilot school's layout is appropriate for a green fence installation, with a playground located at the front and facing a two-lane street. The school's building was constructed in the 1940s and is listed as a cultural heritage site for its neocolonial style. Children of 4–5 years of age use the playground, whilst cars, public transport, and some heavy vehicles produce heavy traffic on the front street and a main road within 60 m of the school. We planted along the perimeter fence between the playground and adjacent sidewalk. The green fence comprises a row of the *Hedera helix* climber growing vertically over railings; two species of bamboo (*Phyllostachys aurea* and *Bambusa multiplex*) to create a second vertical layer behind this, and shorter plants (17 species) towards the playground's interior. Two years after initial installation, the plants were 2.00 m high and 0.50 m wide and were expected to grow in width as the green fence matures. Figure 4 shows illustrative photographs of the green fence in the school playground at earlier stages. We initially attempted to use basic AQ monitoring mobile equipment to measure the green fence efficacy. However, we had to abandon this strategy due to the lack of local staff available to carry out the monitoring amidst COVID-19 restrictions and an extended school lockdown. Instead, we

opted for fixed devices (diffusion tubes) to monitor nitrogen dioxide over approximately three months (three cycles of 21 days each), with support from a local expert who advised us to use this method.

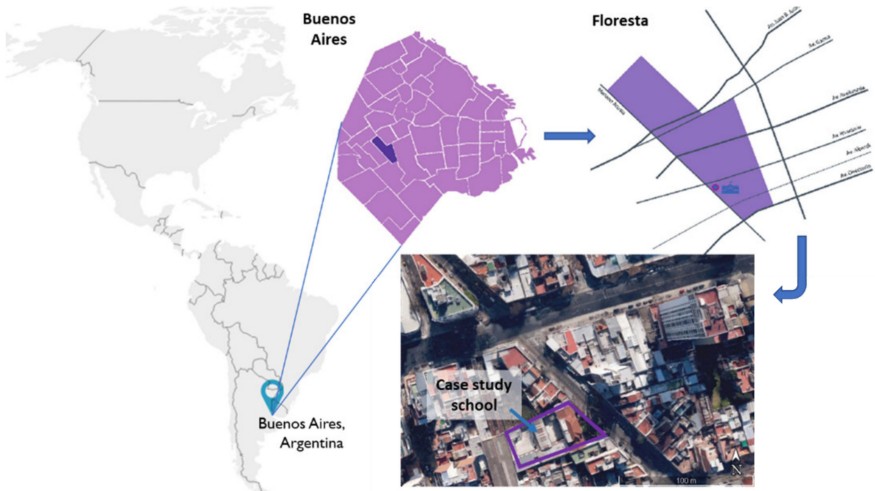

**Figure 3.** Location of case study school where the GF-BA project took place.

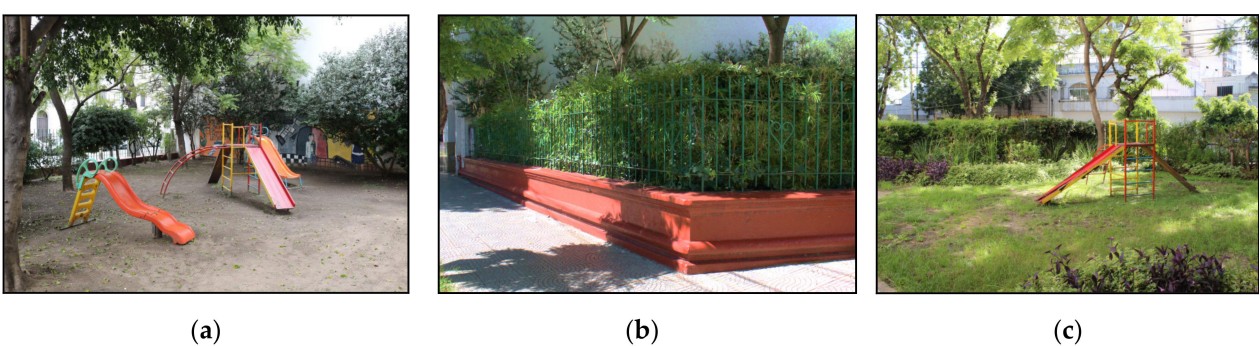

(**a**) (**b**) (**c**)

**Figure 4.** Green barrier in case study school (**a**) before the installation of the green barrier; and 1.2 years after its implementation, from (**b**) street view and (**c**) playground view. Source: Author's own images.

We used this two-year action research experience to develop an explanatory case study on potential solutions for the barriers to implementation that may be preventing broader use of green fences and other GI formats in Buenos Aires. Case studies provide context-specific analysis and thus constitute an appropriate format to capture and present results from action research [32]. With a processual concern on 'how' questions, the case study methodology develops an in-depth description of social phenomena [35]. Case studies are particularly appropriate when the focus is on contemporary questions within complex real-life contexts [35] (p. 18). An 'explanatory' case study provides insights towards theory-building on causal links occurring around a tightly bound process or a 'specific, complex and functioning thing' [36] (p. 175)—in this case the barriers to implementation that we identified and the actions that we took to overcome them. We used a two-step approach to first identify and classify implementation barriers related to the city's policy and social contexts and then reflect on how we cleared them by taking specific actions that redefined the GF-BA project as a GI4AQ+ intervention. The case study was also informed by results from two sets of qualitative methods, namely stakeholder and expert interviews and project members' narratives.

2.2.1. Stakeholder and Expert Interviews—Barriers to GI implementation in Buenos Aires

From the outset, it was clear that the challenges that we were facing were not unique to our project bur rather embedded in broader barriers to GI provisioning in Buenos Aires. Thus, whilst our unit of analysis was the GF-BA implementation process, we carried out interviews with stakeholders and expert informants that helped us interpret the specific issues that our green fence installation faced in the local context where AQ-focused NbS initiatives were underway despite a highly sympathetic policy discourse. We conducted the semi-structured interviews (n = 12) between November 2019 and August 2020. Some of these took place online due to COVID-19 pandemic disruptions. The list of interviewees comprises representatives from different city government units (Ministry of Education, Ministry of Urban Development, Environmental Protection Agency, and City Council), relevant social movements (environmental NGOs and grassroots gardeners' co-operatives), project collaborators from the University of Buenos Aires (staff and students), and landscape volunteers. Thematic analysis was used for interview transcripts alongside policy documents obtained during the literature review stage [37]. The method consisted of identifying codes, which could contribute to answering the research questions [38] (p. 4), and condensing them into meaningful themes based on code similarity. Using the NVivo software package, data were classified into seven categories, derived from the barriers to GI literature review to find common and emerging themes [39].

2.2.2. Project Narratives—The GF-BA Project Attributes

To better understand how implementation was achieved, overcoming the multiple barriers encountered, we constructed a post hoc collective project narrative on the process. A four-member international team had previously conducted participant observation whilst partaking in various activities and interacting with multiple stakeholders, gatekeepers, and contributors. A fifth researcher interviewed them to articulate a joint narrative, recognising that individual experiences would be influenced by respective fields of expertise (landscape planning, environmental science, and social science), positionalities (three female researchers at multiple career stages and one mid-career male researcher), and availability. Three of the members were based in the United Kingdom and travelled to engage in key project events, including the project launch, design workshops, and planting days. The fourth researcher was locally based in Buenos Aires and carried out continuous project management. The analysis focused on the identification of project attributes and associated actions that were either critical to overcoming various implementation barriers towards pilot delivery or outcome enhancement.

## 3. Results

### 3.1. Barriers to Green Infrastructure Implementation

This section presents findings from our review of the general GI literature. These studies analyse implementation challenges and the strategies to overcome them, and provide insight on how green fences research could turn towards questions of effective real-life implementation to realise the pollution-abating benefits calculated through computer models. Using individual case studies and GI compendiums, researchers have identified the most common barriers to implementation, and there is a growing literature on learning from what has worked in collaborative experiments with NbS [26,40,41]. Table 1 summarises findings on barriers across seven categories: (1) institutional and organisational, (2) engagement and coordination, (3) political, (4) socio-cultural, (5) built environment and natural landscape, (6) knowledge, and (7) financial. Extant studies show that lessons are drawn from situated projects often implemented in complex contexts [16] (pp. 8–10), such as our case in Buenos Aires.

**Table 1.** Barriers to green infrastructure identified in the literature, per category.

| Categories | Barriers | References |
|---|---|---|
| 1. Institutional and organisational | Sectoral silos<br>Issues with partnership working<br>Issues with multi-disciplinarity<br>Staff turnover | [27,40,42–44] |
| 2. Engagement and coordination | Unclear responsibilities and ownership<br>Disconnection between short-term and long-term focus<br>Lack of engagement by partners and stakeholders | [27,40,42,45–47] |
| 3. Political | Lack of political will<br>Lack of supportive legislation and regulatory frameworks<br>Lack of law enforcement | [40,44,47,48] |
| 4. Socio-cultural | Path dependency<br>Lack of awareness of multifunctionality and benefits of GI<br>Public preferences and perception<br>Broad societal cultural barriers | [40,42,45,48,49] |
| 5. Built environment and natural landscape | Design limitations<br>Technical difficulties<br>Public and private land ownership | [40,43,47–49] |
| 6. Knowledge | Lack of knowledge of implementation processes<br>Uncertainty as to GI impacts<br>Lack of technical guidance for maintenance and monitoring | [40,42,44–47] |
| 7. Financial | Lack of financial incentives<br>Lack of financial resources | [40,42,46,47,49] |

### 3.1.1. Institutional and Organisational

Green infrastructure projects are multidisciplinary and, as such, entail challenges of leadership and collaboration within and among the institutions involved. Each institution, department, or stakeholder operates on its own terms (e.g., agenda, timeframe, and values). When the different actors in a GI project work according to their own terms, or have goals incompatible with other actors, this is generally referred to as 'sectoral silos' in the literature [40]. Sectoral silos cause institutional fragmentation, which complicates and challenges the progress and success of GI interventions, the common goal. Additionally, leadership and monitoring of GI projects may be hindered by an institution's internal governance issues or staff turnover.

### 3.1.2. Engagement and Coordination

Long-term engagement of all parties involved is a major barrier to GI projects. Outcomes and impacts are usually measurable after a sustained period, in which the role that each actor plays in the long term should be clear. Poor communication among actors causes misunderstanding of project ownership or lack of engagement, which is crucial for the monitoring and maintenance of GI. For instance, when local communities in South Africa claimed a natural park and wetland as part of job loss compensation, uncertainty regarding actual ownership of the area led to neglecting maintenance activities, creating a fire hazard to their farms [50]. Without stakeholder engagement over time, the multiple benefits of GI are potentially undermined.

### 3.1.3. Political

Transitioning from grey to green practices is related to political will. Lack of government support translates into lack of policy, legislation, and regulatory pathways to GI mainstreaming or to poor enforcement of laws and regulations relating to green practices. Lack of political will manifests as prioritising other pressing issues such as poverty or unemployment, aversion towards change, or simply not giving importance to GI as a multifunctional solution. For instance, Johns [48] identified that the persistent prioritising of grey infrastructure over the use of GI in storm water management in Toronto is a significant hurdle perceived as attributable to poor political will.

### 3.1.4. Socio-Cultural

The values of local cultures and gatekeeper institutions where GI interventions take place are important and may pose some socio-cultural barriers. Some of these barriers come from the lack of awareness of the environmental, social, and economic benefits of GI or from a lack of knowledge of successful projects. Past positive and negative experiences with GI at different levels (government, private practice, residents) may create bias towards their uptake or rejection, which is referred to as 'path dependency' in the literature [47].

### 3.1.5. Built Environment and Natural Landscape

The physical characteristics of the place where GI is to be implemented constrain the design and may even impede its development. Firstly, land ownership determines the activities that are possible in a place as well as the gatekeepers to liaise with. Secondly, the landscape morphology, available space, and built-up infrastructure pose technical difficulties in developing GI and dictate its design. Equally important, the availability and structure of adequate vegetation for GI pose technical barriers and constrain the design. For example, Li et al. [51] argue that the current use of a single 'sponge city model' for solving flooding through GI in China may not be successful due to the diverse geographical conditions of the pilot cities.

### 3.1.6. Knowledge

There is little guidance on the development, monitoring, and maintenance of GI, and it is challenging to provide standards because the system is alive (comprising, e.g., vegetation, soil, water) and context-dependent. Therefore, the lack of knowledge of GI practicalities and uncertainty as to its impacts remain as significant hurdles to implementation. Additionally, institutions may not have the capacity or expertise required to develop GI projects as, in some geographical contexts, GI innovation has primarily remained in academia or at pilot project level.

### 3.1.7. Financial

Financial barriers are related to the lack of dedicated resources for the potentially high up-front costs of GI investments as well as long-term maintenance costs. Moreover, a lack of financial incentives is a significant inhibitor to mainstreaming in some countries.

These barriers have by no means deterred cities from undertaking numerous GI and NbS projects. In fact, there is increasing literature documenting lessons learned from successfully implemented schemes. For example, in the European context, where NbS are often initiated by local governments, Frantzeskaki [26] points out that the development of trust between 'city and citizens' and collaborative governance among local government and other actors are important enablers. Yet, this may not translate easily to Latin American contexts with radically different state–community relations [52]. Dobbs et al. [22] calls for reflective practices that leverage global lessons to complement rather than replace existing local initiatives. These require consideration of socioeconomic inequalities and weak governance in the region. Therefore, we adopted an action research approach whereby implementation was collaborative yet researcher-initiated. We built a pilot green fence in Buenos Aires to investigate NbS contribution to addressing local AQ challenges.

### 3.2. Air Quality and Green Infrastructure Approaches in Buenos Aires, Argentina

The Autonomous City of Buenos Aires (hereafter referred to as Buenos Aires or BA) is the political capital of Argentina and the economic and administrative core of the country's largest urban region. Large motorised-traffic flows constitute the main contributor to air pollution [53], as the city's resident population of 3 million doubles during daytime hours. Large, radial boulevards channel traffic to central districts, compounding the problem along street canyons, which present little to no consideration for buffering traffic-related air pollutants, even if provisioned with some level of street-tree planting. Scientific studies have been raising AQ concerns for two decades [54–58]. These studies have correlated concentrations of motorised-vehicle traffic emissions with concentrations of particulate matter ($PM_{2.5}$) [55] and with harmful gases ($NO_2$) [54]. Despite this association, the automotive fleet in BA is on the rise, with 1,548,383 vehicles by June 2020, a 35% increase over 2010 [59], and with no governmental restrictions of traffic flows.

Levels of air pollution in BA sometimes exceed the parameters recommended by the WHO, and at times even the laxer local regulation, as summarised in Table 2. Whilst, according to limited government monitoring, the official maxima are not exceeded, local and periodic exceedances are likely to cause risks to public health. Meteorological factors such as continental winds influence exceedance occurrence, with winds coming from the northwest and southwest causing the largest exceedances in the city [58]. A non-peer-reviewed study commissioned by Greenpeace Argentina [60] to monitor AQ around schools and children's hospitals showed that at more than 40% of the sampling locations, $NO_2$ emissions exceeded the previous WHO limit (2005), and that 14 out of 17 schools had $PM_{2.5}$ concentrations above the same guidelines [61]; under the new WHO guidelines (2021) [62], schools have even greater air pollution exceedances.

**Table 2.** Summary of air pollutant limit guidelines and reported average concentrations in Buenos Aires.

| Type | Reference | $PM_{2.5}$ | $PM_{10}$ | $NO_2$ | $O_3$ |
|---|---|---|---|---|---|
| Guidance | Legislature of Buenos Aires limits [63] | 65 µg/m$^3$–24 h 15 µg/m$^3$–annual (mean) | 150 µg/m$^3$–24 h 50 µg/m$^3$–annual (mean) | 100 µg/m$^3$–annual (mean) | 235 µg/m$^3$–1 h 157 µg/m$^3$–8 h (mean) |
| | WHO limits (2005) [61] | 25 µg/m$^3$–24 h 10 µg/m$^3$–annual (mean) | 50 µg/m$^3$–24 h 20 µg/m$^3$–annual (mean) | 200 µg/m$^3$–24 h 40 µg/m$^3$–annual (mean) | 100 µg/m$^3$–8 h (mean) |
| | WHO limits (2021) [62] | 15 µg/m$^3$–24 h 5 µg/m$^3$–annual (mean) | 45 µg/m$^3$–24 h 15 µg/m$^3$–annual (mean) | 25 µg/m$^3$–24 h 10 µg/m$^3$–annual (mean) | 100 µg/m$^3$–8 h (mean) |
| Reported in scientific study | Bogo et al. [55] | Summer: 41 ± 9 µg/m$^3$ Winter: 33 ± 5 µg/m$^3$ (mean ± sd) | - | - | - |
| | Reich et al. [57] | - | - | 32.7 ± 0.56 µg/m$^3$ (mean ± SE) 1.5 months | 15.88 ± 0.39 µg/m$^3$ (mean ± SE) 1.5 months |
| | Arkouli et al. [56] | 15 µg/m$^3$ in one year | 34 µg/m$^3$ in one year | - | - |
| | Pineda Rojas et al. [58] | - | 29.33 µg/m$^3$ Daily mean for 2008–2010 period | 38.23 µg/m$^3$ Daily mean for 2008–2010 period | - |

The city's AQ policy framework is characterised as incipient. Early adoption of environmental management in the 1960s and 1970s was truncated by Argentina's military coup. National policies only reincorporated democratic engagement with the environment in the late 1980s in the wake of the Brundtland Report [64] and the international drive towards sustainable development. Momentum was gained in the post-Kyoto context of the early 21st century, again with the prominent intervention of international actors such

as the Clinton Foundation, who emphasised the promotion of policies geared to reduce greenhouse gas (GHG) emissions. As a result, the existing monitoring infrastructure is inadequate to detect AQ indicators of critical importance for human health (such as $PM_{2.5}$) and provide reliable data on local exceedances. The General Directive of Environmental Control of the city's Environmental Protection Agency (APrA) has only three operational fixed air-quality monitors within the city, measuring carbon monoxide, $NO_2$, and $PM_{10}$; $PM_{2.5}$ is not monitored at all. Furthermore, efforts towards international compliance are dissociated from local concerns, and there is little integration across policy areas to address crucial AQ challenges in a context of limited government resources and multiple socio-environmental issues.

On the other hand, the BA government acknowledges the importance of urban greening. They have put forward an ambitious plan of GI provisioning as part of the city's commitment to the C40 international cities network of climate change leadership. The local citizenry also mobilises around greenery, with the defence of public space as one of the priorities for both formal NGOs and neighbourhood groups [65]. The city government launched the ambitious Buenos Aires Verde (BAV) programme in 2014. A twenty-year plan, BAV aims to increase public green space in the city, improve local accessibility, and promote pedestrian travel. It relies on new green spaces such as green corridors, green roofs, green highways, urban tree planting, and macro-manzanas (mega-blocks where the built-up infrastructure is demolished and transformed into public green space and amenities). Additional goals include reducing energy consumption, mitigating urban heat island effects, preventing flooding, and ensuring that every resident can access a green space no further than 350 metres from their residence [66]. Yet, in this strategic GI focus on climate change mitigation and adaptation, which BA shares with other municipalities in the Greater Buenos Aires urban region, NbS remains secondary and ancillary to actions related to energy saving and waste reduction [67]. Murgida et al. [68] also point out that the policy framing of emissions abatement has not been properly linked with questions of human health and wellbeing. In Dadon et al.'s [67] taxonomic survey of the development of GI, none of the typologies (urban conservation areas, parks, green roofs, pocket farms) addressed air pollution directly, and in some cases (such as tree-lined streets), they may increase local concentrations of pollutants by preventing quick dissipation [10].

The low levels and uneven distribution of public green space in Buenos Aires is of concern for both strategic NbS and the everyday life of city residents. Buenos Aires reports a stock of green space covering 1827 ha across 1256 sites, with various configurations, including regional, neighbourhood, and pocket parks, squares of varying sizes, planters, and gardens on the median strip of city streets and avenues [69]. This equates to less than 10% of the city's land surface area and results in a ratio of approximately 6 $m^2$ per capita, well below the minimum recommended international standard (10 $m^2$ per capita). Furthermore, some of these green areas are not accessible to citizens, such as planters and gardens on the median strip of avenues, reducing the green ratio to at least 4.59 $m^2$ per capita [70]. Public green space is unevenly distributed, with most of the stock contained within three regional parks located in the city's southern, northern, and coastal edges. In residential areas close to the city's central business district, the formerly industrial south, and densely developed neighbourhoods along western corridors, provision is below average: less than 1 $m^2$ per capita in some districts [71]. The Bunge and Born Foundation's Atlas of Green Spaces estimates that 350,000 residents (12.5% of total population) lack direct access (within a ten-minute walk) to a public green space of at least half a hectare [72]. The advocacy organisation Asuntos del Sur [69] points out that the number of city trees (currently reported at 470,000) would need to more than double to 1 million for Buenos Aires to meet the ratio of one tree for every three residents as recommended by the WHO.

Localised deficits in public open space generate an unmet demand, which in turn results in patterns of overuse and a contested design process for new site provision. In this fraught context, NbS may be perceived as competing with, rather than acting in synergy with, citizens' demands for outdoor exercise and recreational opportunities. It is

no longer only green space advocates who animate campaigns for new parks and green spaces. Since the 2001 economic and political crisis in Argentina, a renewed wave of neighbourhood activism has translated into these planning processes becoming more complex and contested [65]. Whilst all actors criticise the government for the ongoing privatisation of public land, with large vacant lots and interstitial lands being repurposed for real estate development instead of much needed green spaces [73], there is a mismatch between the relatively high social appreciation of existing parks [74] and their low levels of ecosystem services. Civeira et al. [75] estimate the latter based on a ratio of biomass to population density. The recently opened Manzana 66 (Plaza de los Vecinos), in Balvanera, provides an example of such tensions [76]. The results of grassroots neighbourhood efforts, the block has undergone regeneration as a neighbourhood park and primary school. Participatory design processes have resulted in a combination of green and activity sectors but with an overall low proportion of vegetation and exercise areas, with the school directly exposed to thoroughfares with high levels of vehicular traffic. Furthermore, residents raise concerns of deteriorating vegetation in planted areas due to lack of irrigation and upkeep [77].

In summary, BA has a policy framework in place for the creation of NbS. This framework, however, centres on climate change mitigation and adaptation and lacks any reference to how GI could help abate air pollution and protect human health and wellbeing. Accordingly, green fences do not feature in the city's policy toolbox. The undersupply of public green space, and strains on existing spaces, compound the challenges, which calls for innovation in the evidence-based promotion of NbS and a broader perspective on how and where a GI4AQ+ strategy could be feasible and useful.

### 3.3. Barriers to GI4AQ+ Implementation in Buenos Aires

The GF-BA encountered all the barriers commonly identified in the GI literature but also five emergent context-specific barriers. Table 3 provides a summary of both barrier types. The following sub-sections elaborate on the latter.

**Table 3.** Summary of identified barriers to green fences in Buenos Aires and links with existing literature. Orange colour denotes GI4AQ+ in Buenos Aires barriers aligned with previous research, and green colour denotes emergent barriers.

| Barrier Categories | Barriers Identified in Buenos Aires (GI4AQ+) | Linked to Previous Research | Emergent Barrier |
|---|---|---|---|
| 1. Institutional and organisational | 1.1 Multi-disciplinary integration challenges | ███ | |
| | 1.2 Poor inter-governmental and inter-departmental integration | ███ | |
| | 1.3 School governance challenges | ███ | |
| 2. Engagement and coordination | 2.1 Lack of clear pathways to formally engage with government | | ███ |
| | 2.2 Poor diagnosis and communication of AQ | | ███ |
| | 2.3 Gatekeeper institutions engagement | ███ | |
| | 2.4 School community engagement | ███ | |
| 3. Political | 3.1 Unsupportive policies and legal frameworks | ███ | |
| 4. Socio-cultural | 4.1 Lacking AQ awareness | | ███ |
| | 4.2 Predisposition due to previous experiences | ███ | |
| 5. Built environment and natural landscape | 5.1 Design limitations | ███ | |
| | 5.2 Plant availability | | ███ |
| 6. Knowledge | 6.1 Uncertainty regarding GI's effectiveness | ███ | |
| 7. Financial | 7.1 Access to AQ monitoring equipment | | ███ |
| | 7.2 Funding expenditure | ███ | |

### 3.3.1. Institutional and Organisational

Poor coordination between governmental units and other actors in the project produced multiple challenges. An interviewee recalled the GF-BA project team requesting the felling of a Ficus tree from the playground to prevent risks to children, increase sunshine, and enable greater flexibility in selection of plants. The school did not have the resources to undertake and/or pay for this. Furthermore, the two government authorities involved (the Ministry of Education and the Department of Communal Works and Maintenance) could not successfully negotiate the tree removal. Eventually, the project team succeeded in finding additional funds and completed the removal. Despite efforts by all parties involved, this event highlights the lack of clarity concerning ownership, responsibility, and relationships among government units. Additionally, challenges to initiate and maintain school community engagement resulted from how the city's education system is organised, with staff rotation and head teachers responsible for multiple schools. In our case, the head teacher oversees not only the pilot school, but also several other infant schools in the neighbourhood. Thus, her office is not located on site. This situation alone complicated the organisation of regular meetings to involve her in the project.

### 3.3.2. Engagement and Coordination

Engaging stakeholders and securing gatekeeper institutions is a common challenge for GI projects. However, we identified two context-specific barriers. Firstly, there was no clear knowledge of the pathways or means for civil society to engage directly with the government and propose GI projects expediently. This applied to both the executive and legislative branches of government. Some interviewees mentioned that they had to engage with the legislature and multiple ministries through informal mechanisms, relying on pre-existing connections. Our experience was similar. A contact at the Directorate of Urban Anthropology directed us to the relevant unit, the GS Programme from the Ministry of Education, who became the project's gatekeeper. GS promotes sustainable management and environmental education in schools. Yet, formalising a collaborative project was arduous and time-consuming. Despite our project's clear alignment with the unit's remit, an extensive and complex paperwork process of permits was required. Even as the collaboration consolidated over time and the programme's benefits for the pilot school became clearer, GS required the project not to contact school staff directly, as part of their mandate is not to interfere with curricular activities and create burdens on the already limited staff time. In consequence, co-design and co-production activities had to be severely curtailed.

Secondly, whereas engagement of social stakeholders in the GI planning process is often suboptimal due to funding and time limitations, for the GF-BA project, multiple factors compounded the gatekeeper-related barriers mentioned above. We faced pressure to install the fence first and demonstrate its benefits later, skirting around a participatory discussion on air pollution that might have led to community buy-in to site-specific remediation action for the schoolyard. We discovered that the citywide shortcomings in AQ monitoring and reporting were reflected in a local lack of concern with ambient air pollution, despite the school's direct exposure to heavy traffic. Without site- and neighbourhood-specific data, AQ did not prove to be an issue that would mobilise local stakeholders initially. We had to recruit unpaid volunteers more broadly, including landscape activists and various supporters, who contributed to planting without formal project roles. Expert interviewees corroborated our experience, describing the city's AQ data (provided by APrA's webpage) as 'raw' and 'difficult to visualise'. There are not only gaps in the variables reported (with the critical omission of $PM_{2.5}$) but also missing explanations of how high levels may affect human health.

### 3.3.3. Political

Several interviewees decried a lack of green policies, perceiving government as unsupportive of measures to address the city's environmental challenges. In fact, the mismatch

between the city government's green plans and its actual commitment to concrete actions was remarkable. For instance, an interviewee pointed out that little attention is given to the remediation that street trees and other GI could effect on the city's heat island problem. The lack of supportive policy is a widely identified barrier in the GI and NbS literature. In Argentina, it might be related to prioritising immediate actions to solve the economic recession that the country has experienced in the past three years. It might also be related to real estate interests bypassing urban planning by the government, causing the loss of green space to housing. García-Jerez [78] refers to this as 'urban extractivism' and points out that it is an emerging problem in many Latin American cities.

### 3.3.4. Socio-Cultural

Lack of public awareness of air pollution's effects on human health constitutes a major barrier to the promotion of AQ-related GI initiatives in Buenos Aires. The lack of pre-existing NbS interventions to shape the agenda constitutes a contributing factor already identified in the international literature. Our findings also indicate that three case-specific factors contribute to perceptions that AQ does not constitute a major policy challenge for the city. Firstly, the governmental reporting (AprA) that the city (partially) complies with local and international regulations on 'environmental quality' does not consider pollution hotspots, which the public monitoring system is also unable to identify and document. Secondly, civil society campaigns and awareness-raising actions are few and far between. An interviewee reported major governmental pushback to their 'alarmism' and 'scaremongering regarding what is really a non-issue'. Thirdly, unlike other Latin American large cities such as Santiago de Chile, Mexico City, and Bogotá, air pollution is not sensorially evident citywide for extended periods of time, with numerous cases of poor health symptoms (headaches, eye irritation, coughs) reported [79,80]. Without educational and awareness campaigns, long-term effects and chronic health implications are less evident.

### 3.3.5. Built Environment and Natural Landscape

As in other urban environments, the introduction of GI in Buenos Aires schools faces design limitations. Some interviewees highlighted that the 'cultural heritage' status of certain school buildings with colonial features prohibits interventions that may change their appearance. This was the reason for the refusal of permission to install an irrigation system for the GF-BA. Other schools do not have suitable planting space. Planning and building code constraints are a common barrier in the literature, highlighting the importance of adapting the design to the specific contexts and proposing alternative GI typologies.

The availability of suitable plants for GI4AQ+ was identified as a barrier in Buenos Aires, which is not discussed in previous studies conducted elsewhere. Plant selection relies upon scientific literature primarily generated in Europe, the US, Australia, Japan, and China. Most of the species investigated in these countries are different to the ones commercially available in Buenos Aires, or they are considered a specialty in the local market, which increases costs. Additionally, there is resistance to the use of species that are not local and native to the region, as an interest in preserving native species has emerged and gained momentum across Argentina [81].

### 3.3.6. Knowledge

Our findings indicate scepticism regarding the efficacy of GI schemes among policymakers and stakeholders in Buenos Aires. This is consistent with findings from elsewhere reported in the extant literature and is compounded by the lack of awareness and data on air pollution. Even supportive policymakers requested primary evidence that green fences would work as intended once the pilot green fence was installed. In fact, this was a requirement before proceeding with implementation in additional schools (see Figure 2 for details). The process of AQ monitoring and data collection encountered difficulties related to the lack of familiarity with GI4AQ interventions among specialists and the need

for them to adapt their techniques and research design to test the green fence's efficacy. Our first monitoring campaign with diffusion tubes indicated lower $NO_2$ concentrations within the green fence's perimeter and possibly accelerated pollutant dispersion in comparison to a control pair of diffusion tubes installed on a fence-less side of the school (see Supplementary Materials). However, we decided to carry out a secondary campaign with a larger number of tubes on both the intervention and control sides to obtain a clearer indication of diffusion rates, and have yet to implement a $PM_{2.5}$ monitoring system suitable to the site conditions and local technical capacity. Whereas socio-ecological co-benefits proved easier to document, with a very positive response by the school community to the greenery that was installed, knowledge about these was also limited before the intervention, and detailed educational activities were required to familiarise various schoolyard users with the green fence beyond their initial sensorial reaction. Evidence about local plant species with bioremediation potential and AQ benefits is scarce. Yet, we found that exploring the potential use of local species, on which the literature is scant, would require further efforts and a large research infrastructure. This includes plant science labs, high-tech microscopes, and trained personnel, none of which are readily available currently.

### 3.3.7. Financial

A lack of clear financial incentives and dedicated resources is a frequent challenge to GI development, which we also identified in Buenos Aires. Whilst the city has a budget for planting in public spaces, this is not explicitly linked to potential socio-environmental benefits through landscape-level planning. Furthermore, evidencing air pollution beyond what the city reports required costly imported equipment, which constituted an emergent financial barrier not discussed in previous studies. We had options such as low-cost sensors, but these involved technical (internet and power in situ) and resource hurdles (human resources to regularly monitor in situ for a sufficient period). Therefore, establishing a reliable AQ monitoring system for the GF-BA project became a challenge that actually delayed the pilot's implementation. Paradoxically, our government partners required the project to evidence the green fence's effectiveness in terms of AQ improvements as a condition for continuing engagement, even when reliable city data was not readily available to benchmark the intervention.

### 3.4. The GF-BA Project Attributes: Overcoming Barriers and Producing Benefits

This section presents the results of the researcher narratives to provide an account of how we were able to successfully complete the GF-BA pilot project, overcoming the multiple barriers to implementation presented above. Furthermore, the (many unanticipated) actions that we needed to take led the project team to examine the project's core attributes, sustain the work through the implementation challenges that we faced, and eventually reformulate an expanded model of GI4AQ+ learning from the experience.

The GF-BA project features three salient attributes. The pilot's completion relied on (a) a relational commitment to the urban environment by multiple and diverse participants; (b) activities inspired by cross-boundary experimentation and innovation; and (c) the support of a horizontalist transnational research collaboration. These three project attributes informed multiple actions that enabled the pilot to overcome implementation barriers. Figure 5 shows the relationship between these 'actions of critical importance' and barrier categories. The actions were critical because, without them, we could not have kept the project advancing through its implementation stages, as some barriers would have been unsolvable. Other ancillary actions were also taken to enhance the green fence, including ongoing communication among project leads and efforts to publish and disseminate across multiple formats, but these are not shown here, for explanatory clarity. The figure shows that all three project attributes and their corresponding actions were a conduit to overcoming engagement and coordination barriers. Engaging with the BA government early on, finding the project's gatekeepers, and gaining acceptance to work with them relied on the relational, experimental, and transnational nature of the GF-BA

project. Moreover, the relational commitment to the urban environment highly influenced all the barriers associated with social interactions. Cross-boundary experimentation and innovation helped to solve not only social barriers endemic to interdisciplinary efforts but also technical barriers happening on the ground. Finally, the horizontalist transnational research collaboration was highly important in allowing us to access funding and enabling design of the green fence, overcoming built environment, natural landscape, knowledge and financial barriers.

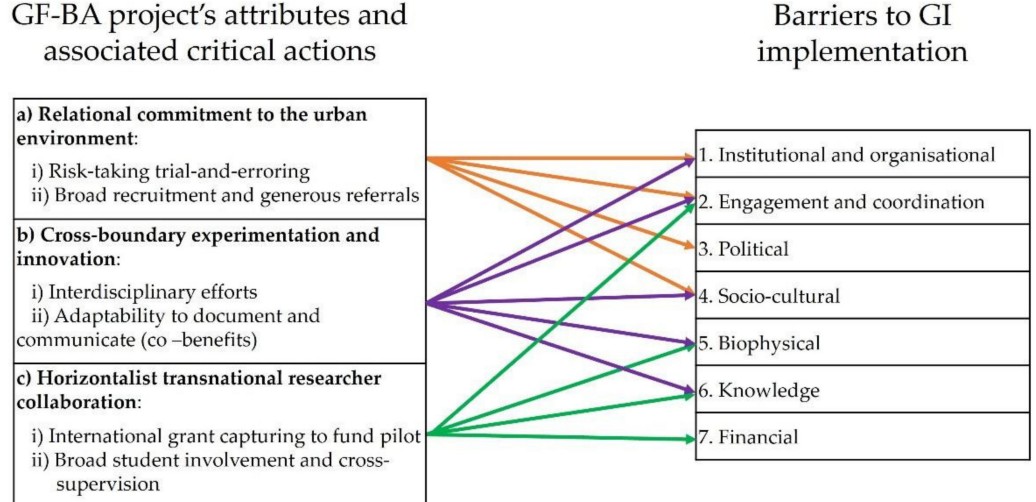

**Figure 5.** Relationship between barriers to GI implementation and the GF-BA project attributes and associated critical actions.

From the beginning, the project brought together diverse people committed to help improve the urban environment in Buenos Aires. Contributors worked in either higher education, government, NGOs, or the private sector. Project members and initial collaborators were willing to take professional risks and/or contribute personal time to a project with uncertain local outcomes. Enthusiastic responses to our project secured us broad recruitment and generous referrals that opened doors to various local and national institutions. Occasionally, we relied on contacts obtained through previous environmental actions. Once installed, the green fence itself helped us recruit further collaborators and supporters. The school's groundkeeper volunteered to water the plants over the summer recess month, which was essential for maintenance, given that we were not allowed to install an automated system. Acting on her own initiative, a neighbour started commenting on social media on the project's positive local impact. Whilst voluntarism may appear as a shaky foundation for a project's success, the many enthusiastic contributions that our green fence proposal received in Buenos Aires gives us grounds to believe that willingness to improve the urban environment may also be found elsewhere for similar catalyst projects focused on place-specific actions.

In the face of multiple local uncertainties and lack of precedent, and a globally evolving field of GI applied to AQ challenges, it was clear to us that the GF-BA project needed to be designed as an experimental pilot project. The challenge resided not only in ascertaining what type of fence design (including plant species selection) would produce some form of demonstrable AQ site improvements but also in how we could communicate these benefits effectively to secure social and policy impact towards broader green fence adoption in the city. This spearheaded a truly interdisciplinary effort, which was not limited to the initial collaboration between environmental scientists (including air quality specialists and a material science expert), landscape designers, and social scientists specialising in urban and environmental policy in Latin America. Our activities soon broadened to include local and international experts in epidemiology and public health, ecology and biodiversity, the popular economy (in issues pertaining to those workers who may be involved in

the construction of green fences and other types of GI), environmental and nature-based education, and participatory methods with school-aged children. The emphasis that we placed in dissemination and knowledge exchange, with frequent seminars by invited experts and a programme of workshops with the school community, helped the project gain visibility with research and advocacy communities, but also provided us with feedback on how to adapt our programme of activities and communication strategy regarding early results. This strategy was needed to secure ongoing support from governmental gatekeepers and engagement from social stakeholders.

The collaboration between UK and Argentine researchers at the core of the GF-BA project (see Figure 2) was instrumental to the successful pilot completion. Both partners contributed essential, mutually complementary components. The former's participation secured British research funding required to jump-start the project and then build the pilot in the face of no initial local budgetary support and the inexistence of bespoke markets to lower the cost of interventions (including imported AQ monitoring equipment and even the relatively high cost of plants). It also provided a track record of previously documented GI4AQ interventions in the UK and the incentive for local government actors to participate in what may be seen as a high-profile international collaboration [73,82]. The latter contributed an intimate knowledge of tacit local policy barriers, the ability to increase the project's visibility among different professional and research circles in the city, and effective communication strategies vis-à-vis different stakeholders with specific preferences and needs. The Buenos Aires-based researcher tied her teaching practice to the project and involved university students who contributed to the fence design, construction, and monitoring. This not only helped mitigate high costs but also contributed to train a future generation of professionals with an understanding and sensibility for the locally untapped potential of urban greenery as GI. The UK side also involved students at multiple levels, including a doctoral project, masters' dissertations, and undergraduate involvement through coursework. Cross-supervision of students helped to build a sense of horizontal transnational collaboration without hierarchical differences, which could have been created by questions of who contributed what resource to the project.

## 4. Discussion and Recommendations

This paper engaged the research and innovation challenges of moving schoolyard green fence interventions from a model of GI4AQ with proven efficacy for site-specific remediation to an urban NbS with broader geographical and functional potential. We emphasised the need to better understand and take stock of implementation challenges. Therefore, lessons from the GF-BA project were used not only to identify barrier categories but also formulate strategies to overcome them and make recommendations for future intervention models. Our GI4AQ+ approach emphasises enhanced green fence designs, with the dual goals of producing multiple socio-environmental co-benefits and increasing the chances of stakeholder buy-in. We argued that in urban contexts of low- and middle-income countries, where air pollution is often not understood as an urgent urban problem or even a policy priority, schoolyard green fences can produce the cultural ecosystem-based service of raising awareness about AQ challenges. If properly designed, they can reduce harmful pollutants on site and visibly contribute to an urban green network to support biodiversity, improve human health and wellbeing, and strengthen the capacity of cities to generate ecosystem-based services.

The GI4AQ+ model moves green fences from a site-specific bioremediation to an urban environmental intervention, building upon the context specificity recommended as a NbS core principle [83]. Our results showed that considering the particularities of an urban landscape, society, and politics are crucial to implement a first pilot fence and avoid assumptions about the local potential of different GI formats. Such in-depth contextual understandings will also be of critical importance if the research and practice of urban NbS are to overcome current biases toward high-income countries, especially in the European context [9], and provide useful insights and workable solutions for the critical urban envi-

ronmental problems for cities in low- and middle-income countries. The paper emphasises that Latin America constitutes a critical geography for a more thorough globalisation of urban NbS, and studies need to develop frameworks that integrate the region's ecological conditions and rich biodiversity with the dynamics of rapid and uneven urbanisation [84] (p. 14), whereby speculative land use change and informal urban expansion without much governmental regulation have resulted in multiple inequalities, including the dramatic decrease of public green areas. Moreover, as Dobbs et al. [22] point out, approaches to urban greening taken in high-income countries may provide guidance and complementarity, but due to the uniqueness and diversity of the region, Latin American studies need specific approaches, avoiding overreliance on foreign research.

The action research and living lab approach of the GF-BA project was highly valuable in teaching us to devise a modified approach more suitable for the local context's needs and priorities. Abating local air pollution seemed less urgent than we expected to multiple local social and governmental actors—even if they remained sympathetic to the overall goal of building a green fence and enhancing the schoolyard environment. Fostering broad-based interdisciplinary conversations about the multiple social and environmental benefits of increasing urban vegetation at the planning stages helped us co-produce a research programme around the intervention's potential co-benefits, which we could then begin to document once the green fence was installed. Whilst these were key to maintaining momentum until we could confirm initial results on AQ (initial readings for $NO_2$ are shown in the Supplementary Materials section), the very process of setting up a monitoring system and communicating its objectives and findings helped us hold otherwise difficult conversations around questions of air pollution in the city and how they may be affecting the school community. Therefore, green fences built following an expanded GI4AQ+ could serve the long-term purpose of environmental education and awareness whilst immediately abating pollution and providing ecosystem services.

We are aware that our GF-BA green fence pilot implementation project only constitutes a first step towards the robust formulation and broad application of a GI4AQ+ model suitable for Buenos Aires, other Latin American cities, and potentially elsewhere. However, this case study contributes an evidence-based experience of GI co-production with policymakers and community buy-in, which faced several challenges but also succeeded as a pilot. Even if developing an urban NbS network of green-fenced schools may not be easily achievable through action research, pilots developed through this approach are useful as both proof of concept and sites for further experimentation and model refinement. When completed, our project was featured on the city government website and helped garner visibility for purposively designed green fences in Buenos Aires. Furthermore, our project has since expanded to include two more participating schools in the metropolitan area (see Figure 2) and develop a programme of site-based experimentation informed by urban living lab approaches [28–30]. This aims to generate further insight into effective green fence design and programming.

Plant selection and species availability/suitability are key items in our ongoing research agenda. Whilst the extant AQ-focused literature relies largely on plants available in China and Europe [85], efforts need to be made to extract generic plant traits from these studies and develop locally suitable plant selection guidelines. Future studies will need to ascertain the possibility of reducing costs and assuaging concerns of working with exotic species whilst not compromising efficacy regarding reduction of exposure to air pollution. Schoolyard use is also a topic for further examination, including how children's play may change after green fences are installed; parents, staff, and neighbourhood reactions to the transformed local environment, and what support may be needed for the incorporation of the NbS to (extra-)curricular contents and activities, must be included in further green interventions in the schoolyard or even the main buildings. Finally, green fences can inform policy-oriented research programmes, even whilst they are being planned and constructed, by steering questions to government officials and expert informants to concrete implemen-

tation questions that may not be captured otherwise, as conversations would be limited to the policy discourse and official frameworks.

## 5. Conclusions

This study probed the potential for mobilising GI to mitigate urban air pollution in Latin America, specifically through the use of green fences in schoolyards in Buenos Aires, Argentina. In this city, the policy response to air pollution levels is barely incipient, with greenhouse gases emissions receiving priority over those pollutants more harmful to the health of local residents. At the same time, the city has an ambitious GI policy framework but lacks effective provisioning, with concerns raised over the diminishing amount of open green space. We argued that this evidence of a low policy priority for AQ, alongside interest in GI, calls for an intervention model whereby green fences should not be limited to site-specific protection from nearby traffic pollution. Instead, the model of GI4AQ+ that we introduced could help raise awareness of the problem and educate citizens about its implications for public health whilst also contributing to their wellbeing through urban vegetation enhancement; it can be incorporated into even more ambitious NbS agendas of urban ecosystem-based services.

We do not assume this to be a simple task. All GI formats face several implementation challenges, including institutional, engagement, political, socio-cultural, geographical, knowledge base, and financial barriers. Through the co-creation process of the GF-BA project and supplementary in-depth interviews, we identified five additional barriers to the development of green fences in Buenos Aires. Three out of the five emergent implementation barriers relate to (i) insufficient and/or inadequate AQ monitoring, data sharing and communication; (ii) lack of citizen awareness of air pollution risks; and (iii) high costs and low availability of equipment for local AQ monitoring. The two other barriers speak to broader challenges to GI development, including (iv) limited clear avenues for co-producing projects with government actors; and (v) limited availability of plant species suitable for phytoremediation interventions. These may not be important issues for NbS initiatives in Europe and North America, but they could prove major hurdles to promote green fences in cities of low- and middle-income countries. Despite all the hurdles and challenges faced during the GF-BA project, the pilot green fence was successfully implemented. Three attributes of the project and their correspondent actions facilitated overcoming the barriers, which relied on the project incorporating 'urban living lab' principles from the NbS approach, mainly through a relational commitment of different people to improve the urban environment regardless of possible risks and setbacks; horizontalist collaboration across disciplinary silos and the North-South divide in support for urban environmental research; and an ethos of scientifically informed experimentation towards social innovation.

The GI4AQ+ model seems to be particularly suitable for cities in development contexts where environmental policy needs consolidation and AQ awareness is lacking. GI4AQ+ may not only reduce exposure to pollutants, but also effectively communicate the importance of improving AQ and practically demonstrate the multiple contributions that GI makes to urban life. This will gradually raise the awareness of students, possibly supported by curricular and non-curricular activities that leverage the green fences' appeal, promote visibility of the issue within the broader school and neighbourhood communities, and provide further eco-systemic co-benefits at multiple scales.

**Supplementary Materials:** The following supporting information can be downloaded at: https://www.mdpi.com/article/10.3390/su14074129/s1, Figure S1: Location of $NO_2$ monitoring devices (diffusion tubes) in relation to the case study school and green fence; Table S1: Results of monthly $NO_2$ concentration monitoring (diffusion tubes) in parts per billion (ppb).

**Author Contributions:** Conceptualisation, M.d.C.R.B., J.M.K., J.A., V.F. and A.J.; methodology, A.J., J.M.K. and V.F.; data curation and formal analysis, M.d.C.R.B. and J.A.; writing—original draft preparation, M.d.C.R.B. and J.A.; writing—review and editing, M.d.C.R.B., J.M.K. and A.J.; supervision, J.M.K. and A.J.; project administration, V.F.; funding acquisition, J.M.K. and A.J. Authorship has been limited to those who have contributed substantially to the work reported. All authors have read and agreed to the published version of the manuscript.

**Funding:** This research was funded by the Grantham Centre for Sustainable Futures, who awarded a PhD studentship to MCRB (BREATHE/RESPIRAR project). The British Academy funded J.M.K., J.A., V.F., and A.J.'s participation in the project (Urban Infrastructures of Wellbeing Scheme—Grant Number: UWB190225). A seed grant from the British Council (Higher Education Links) allowed us to carry out early project activities.

**Institutional Review Board Statement:** The study was conducted in accordance with the University of Sheffield's Research Ethics Policy and approved by the Department of Landscape Architecture Ethics Committee of The University of Sheffield (reference number 031003, approved on 31 October 2019).

**Informed Consent Statement:** Informed consent was obtained from all subjects involved in the study.

**Data Availability Statement:** This study did not use secondary data. The data generated here are available on request from the corresponding author and has been archived and is accessible at https://doi.org/10.15131/shef.data.19434653. The data are not publicly available due to ethical restrictions. Please contact the corresponding author for access or follow the data repository site instructions.

**Acknowledgments:** The authors would like to acknowledge support from the Buenos Aires City Government in the project development, The 'Escuelas Verdes' ('Green Schools') unit from the Ministry of Education was particularly helpful. Support is also acknowledged from the Directorate of Urban Anthropology, the city's Environmental Protection Agency, and the National Congress. We would also like to acknowledge support from academics and students at the University of Buenos Aires, Department of Architecture, Design and Urbanism, Department of Agronomy, and Department of Science. We deeply thank the volunteers, horticultural co-operatives, and the RiveraBA environmental education NGO for their collaboration to the project.

**Conflicts of Interest:** The authors declare no conflict of interest.

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
