# Peer review of "Green Fences for Buenos Aires: Implementing Green Infrastructure for (More than) Air Quality"

_sustainability, doi:10.3390/su14074129_

Round 1

Reviewer 1 Report

Considering the height of the green fence, the results of the AQ monitoring would be really necessary to decree its effectiveness. Please integrate

Author Response

Air quality monitoring (AQM) methods and preliminary results have now been integrated in the manuscript. The rational for the selection of tour AQM strategy is presented in the Methods section, together with considerations regarding the green fence's height. Preliminary AQ results are described in Results (Section 3.3). Moreover, detailed results and a site plan showing the sampling locations have also been incorporated for clarity as Supplementary Material. It is worth adding that the aim of this paper is not to demonstrate AQ efficacy but to assess green fences implementation feasibility.

Reviewer 2 Report

Dear Authors,

First I would like to congratulate you on a very interesting and complete presentation of you manuscript "Green Fences for Buenos Aires: Implementing Green Infrastructure for (more than) Air Quality". It is very clear you have placed a lot of hard work and effort, that will help to improve knowledge of Urban GIs using green fences. You have provided key literature and then present an actual pilot study in Buenos Aires.  This is a very nice complement and is very worthy of publication.

However, I found the structure and framing or you manuscript very distracting and difficult to follow. As a reviewer and reader, I would like to see the manuscript re-structured to follow Introduction, Methods, Results,  Discussion and Conclusion.

Please see the Author instruction on structure these will help set up your manuscript.

You have everything already there but it needs to be re-organized. At present the Introduction is like a extended summary or extended abstract of what you have done.  I found very confusing and hard to follow as a lead in to your study.  This text needs to be re-used in the restructure of you manuscript but not presented as it is. Where as Section 2 reads more like an introduction. 

In the last paragraph of the Introduction following your setup of the gaps and needs for studies on GI towards improved AQ, I would like to see the aims of you manuscript. For example, in this study aims to explores the potential for extending this model of site-specific green infrastructure for air quality (GI4AQ) to cities in low- and middle-income countries beset by air pollution and multiple other socio-environmental challenges. To achieve this we i) review literature on ...., ii) ......, iii) undertake a pilot study on green fences ........ and finally we discuss .....

Or something in this fashion for readers to quickly obtain what you did. this can also be presented in the same format in the abstract.

In the Methods section, I first suggest including a small section (paragraph) to frame your study and present an overview of of what you have done, a figure can also be used. This will provide the reader with a clear direction and support your aim. You can take this from the current Introduction "The paper’s main body is divided into five sections" and even add a figure to show what you do (sell your good work). 

Following it would be useful to have a section for each of the five sections to explain what and how you did. These section heading you have should be repeated and presented in your Results Section. 

Then build your Discussion. 

One final suggestion is to check there are no floating statements that need references. 

  My only concern is with the structure of the paper, your topic, content, findings and conclusion are all worthy of publication. The English is fine. Also I attach the document with some small edits, again it mainly relates to structure of the manuscript.

Keep up the good work you are nearly there!!!

Author Response

Point 1: First I would like to congratulate you on a very interesting and complete presentation of you manuscript "Green Fences for Buenos Aires: Implementing Green Infrastructure for (more than) Air Quality". It is very clear you have placed a lot of hard workand effort, that will help to improve knowledge of Urban GIs using green fences. You have provided key literature and then present an actual pilot study in Buenos Aires. This is a very nice complement and is very worthy of publication.

However, I found the structure and framing or you manuscript very distracting and difficult to follow. As a reviewer and reader, I would like to see the manuscript re-structured to follow Introduction, Methods, Results, Discussion and Conclusion. Please see the Author instruction on structure these will help set up your manuscript.

Response 1: We thank the reviewer for their interest and support to our research. According to their request, we have restructured the manuscript to follow Sustainability’s format, including Introduction, Results, Discussion and Conclusion.

Point 2: You have everything already there but it needs to be reorganized. At present the Introduction is like a extended summary or extended abstract of what you have done. I found very confusing and hard to follow as a lead in to your study. This text needs to be re-used in the restructure of you manuscript but not presented as it is. Where as Section 2 reads more like an introduction. In the last paragraph of the Introduction following your setup of the gaps and needs for studies on GI towards improved AQ, I would like to see the aims of you manuscript. For example, in this study aims to explores the potential for extending this model of site-specific green infrastructure for air quality (GI4AQ) to cities in low- and middle-income countries beset by air pollution and multiple other socio-environmental challenges. To achieve this we i) review literature on ...., ii) ......, iii) undertake a pilot

study on green fences ........ and finally we discuss ..... Or something in this fashion for readers to quickly obtain what you did. this can also be presented in the same format in the abstract.

Response 2: Section 1 has been removed and reused in other sections where the information provided was more relevant and appropriate. The Introduction has been re-written and a clear aim and objectives stated at the end of the section. Additionally, the same objectives  have been followed to re-order/re-write the abstract. The objectivess are:

  1. Review literature on barriers to GI implementation
  2. Review the status of GI and AQ in Buenos Aires, Argentina
  • Undertake a pilot case study on green fence implementation (as GI4AQ+) in a school playground in Buenos Aires, Argentina
  1. Discuss the results of the case study in the light of the current EuroAmerican literature, identifying emergent barriers and solutions.

Point 3: In the Methods section, I first suggest including a small section (paragraph) to frame your study and present an overview of of what you have done, a figure can also be used. This will provide

the reader with a clear direction and support your aim. You can take this from the current Introduction "The paper’s main body is divided into five sections" and even add a figure to show what you do (sell your good work). Following it would be useful to have a section for each of the five

sections to explain what and how you did. These section heading you have should be repeated and presented in your Results Section.

Response 3: The Methods section has been rearranged and the introducing paragraph now provides an overview of the research methods used in the study. Figure 1 has been added for support and clarity of the study’s research design. The Methods section has been divided in further subsections to clarify that we have conducted 1) a literature review, and 2) a green fence pilot study. In accordance with the Methods, the Results section now clearly presents the outcomes from the literature review (sections 3.1 and 3.2) and from the green fence pilot study (sections 3.3 and 3.4).

Point 4: Then build your Discussion. One final suggestion is to check there are no floating statements

that need references. My only concern is with the structure of the paper, your topic, content, findings and conclusion are all worthy of publication. The English is fine. Also I attach the document with some small edits, again it mainly relates to structure of the manuscript. Keep up the good work you are nearly there!!!

Response 4: A Discussion section has been added by rewriting/re-structuring the manuscript, and the Conclusion section has been modified to succintly show the research outcomes. The ‘floating statements’ from former Section 1 have been removed.

Reviewer 3 Report

The article is an interesting overview from the point of view of the implementation of green infrastructure. However, the presentation of the problem is disorganised and needs a thorough reworking. If the authors even refer to the Buenos Aires case study in the title, the presentation of this thread is insufficiently clear in the article.

The layout of the article should clearly define the scope of the problem, but unfortunately this is not the case. All chapters of the article are equal parts, which even repeat each other in terms of meaning. The classic layout of the article, which is characteristic for a journal, is missing, for example: introduction with literature review (despite the presence of the chapter "introduction"), characteristic of the main problem/characteristics of the case study area (its specific conditions), justification of the application of solutions with reference to world literature, discussion and conclusions. The article lacks a clearly focused discussion, which would either confirm the validity of the applied solutions or negate the validity of the solutions used so far in the world. Part of the conclusions should refer to the problem of universality of the "guidelines" in the international context, which should be the result of the discussion.

As it stands, I propose a major revision. The article needs a tidier layout along the lines of the journal articles. The presented issues are interesting, but the authors touch on a lot of threads. If the authors' wish is to present everything in one article, I would suggest shortening and rewriting the content of the issues that are raised. In its present form, the issues raised in such large numbers lose a lot. The article lacks locational reference. No map of the Case Study area with indication of conditions was made. This would also improve the clarity of the article.

Author Response

Point 1: The article is an interesting overview from the point of view of the implementation of green infrastructure. However, the presentation of the problem is disorganised and needs a thorough reworking. If the authors even refer to the Buenos Aires case study in the title, the presentation of this thread isinsufficiently clear in the article.

Response 1: We thank the reviewer for their interest and support to our research. We have worked to improve the structure and clarity of the study, and believe that the changes made reflect the importance of carrying out this research. We have re-written the Introduction to better state the air pollution problem in low- and middle-income countries and the need for innovative measures to protect their vulnerable populations. The Introduction now clearly shows the aim and objectives of the study and their link with Buenos Aires, Argentina. The objectivess are:

  1. Review literature on barriers to GI implementation
  2. Review the status of GI and AQ in Buenos Aires, Argentina
  • Undertake a pilot case study on green fence implementation (as GI4AQ+) in a school playground in Buenos Aires, Argentina
  1. Discuss the results of the case study in the light of the current EuroAmerican literature, identifying emergent barriers and solutions.

Point 2: The layout of the article should clearly define the scope of theproblem, but unfortunately this is not the case. All chapters of thearticle are equal parts, which even repeat each other in terms ofmeaning. The classic layout of the article, which is characteristicfor a journal, is missing, for example: introduction with literaturereview (despite the presence of the chapter "introduction"),characteristic of the main problem/characteristics of the casestudy area (its specific conditions), justification of the applicationof solutions with reference to world literature, discussion andconclusions.

Response 2: We have restructured the manuscript to follow Sustainability’s format, including Introduction, Results, Discussion and Conclusion.

Point 3: The article lacks a clearly focused discussion,which would either confirm the validity of the applied solutions or negate the validity of the solutions used so far in the world. Part of the conclusions should refer to the problem of universality of the "guidelines" in the international context, which should be the result of the discussion.

Response 3: A Discussion section has been added by rewriting/re-structuring the manuscript. It explains the potential of the GI4AQ+ model in Latin America as an urban environmental intervention. Additionally, the Conclusion section has been modified to succintly show the research outcomes.

Point 4: As it stands, I propose a major revision. The article needs a tidier layout along the lines of the journal articles. The presented issues are interesting, but the authors touch on a lot of threads.If the authors' wish is to present everything in one article, I would suggest shortening and rewriting the content of the issues that are raised. In its present form, the issues raised in such large numbers lose a lot.

Response 4: The whole manuscript has been re-strctured/re-written to improve readability, and add clarity to the methods, outcomes and conclusions derived from our research. The Methods section   has been divided in further subsections to clarify that we have conducted 1) a literature review, and 2) a green fence pilot study. In accordance with the Methods, the Results section now clearly presents the outcomes from the literature review (sections 3.1 and 3.2) and from the green fence pilot study (sections 3.3 and 3.4). A Discussion section is now clearly differentiated from the results, leading to a succinct Conclusion section.

Point 5: The article lacks locational reference. No map of the Case Study area with indication of conditions was made. This would also improve the clarity of the article.

Response 5: In the Methods section, a further explanation of the case study school’s location and characteristics has been added. Figure 3 now supports this section and shows the location of the case study school at regional, city and neighborhood levels.

Round 2

Reviewer 2 Report

Dear Authors,

Congratulation on following the reviewers requests and comments. Your revised version has been greatly improved during major revision. I now find the structure of your manuscript easy to follow and inline with the Journals requirements. At the same time you have also  greatly improved the quality and readability of your manuscript.  

I do suggest that you check the English for spelling (for example "schoolward"), correct word usage (for example "grappled" (maybe analyzed)) and concise language (see below), but these are minor issues that you can pick up during the processing of your manuscript.

For instance ". To close these gaps in knowledge gaps, attend to a mitigate serious health threats, and contribute to the broader NbS agenda, the rest of this paper discusses the actual implementation experience of a pioneer green fence and development of the GI4AQ+ model

Well done on a very nice revision.

Author Response

Many thanks for your final considerations. We have revised the manuscript with the word choices you suggested. 

Reviewer 3 Report

The article has definitely gained in clarity after the changes. However, I would like to draw attention to one element.

It is not good practice to create unnumbered paragraphs (see chapter 3.2 Air Quality - approaches in Buenos Aires-Argentina, Institutional and organisational, Engagement and coordination, Political, Socio-cultural etc....). It is not clear what role these headings play, especially since the title of chapter 3.2 does not translate directly into such problem headings. Paragraphs should be numbered to indicate the relationship to the chapter. If the authors make this effort, the article will, in my opinion, be fit for print.

Author Response

Many thanks for your final considerations. We have numbered all sub-headings.